# A Neural Tangent Kernel Approach for Constrained Policy Gradient Reinforcement Learning

## Abstract

This paper presents a constrained policy gradient method where we introduce constraints for safe learning, augmenting the traditional REINFORCE algorithm by taking the following steps. First, we analyze how the agent's policy changes if a new data batch is applied, leading to a nonlinear differential equation system in continuous time (gradient flow). This description of learning dynamics is connected to the neural tangent kernel (NTK) which enables us to evaluate the policy change at arbitrary states. Next, we introduce constraints for action probabilities based on the assumption that there are some environment states where we know how the agent should behave, ensuring safety during learning. Then, we augment the training batch with these states and compute fictitious rewards for them, making the policy obey the constraints with the help of the NTK-based formulation. More specifically, exogenous discounted sum of future rewards (returns) are computed at these constrained state-action pairs such that the policy network satisfies the constraints. Computing the constraining returns is based on solving a system of linear equations (equality constraints) or a constrained quadratic program (inequality constraints). To tackle high-dimensional environments, a dynamic constraint selection methodology is proposed. Simulation results demonstrate that adding constraints (external information) to the learning can improve learning in terms of speed and transparency reasonably if they are selected appropriately.

## 1 Introduction

In reinforcement learning (RL), the agent learns through trial and error. If the agent is deployed to a real-world setting while it has not yet explored the whole state space, unexpected situations may occur, resulting in damage or injury to the agent or the environment. In addition, the agent might waste a significant amount of time exploring irrelevant regions of the state and action spaces. Constrained learning is an intensively studied topic, having close ties to safe learning Garcia & Fernandez (2015); Yang (2019); Berkenkamp et al. (2017); Fisac et al. (2018); Zimmer et al. (2018); Gros et al. (2020). What all the methods mentioned above have in common is some knowledge of the environment. In safety-critical settings, some actions must be constrained, and exploration cannot be done blindly. Han et al. (2008) argues that constrained learning has better generalization performance and a faster convergence rate compared to basic feedforward neural network architectures in supervised learning. Although many works focus on supervised learning, general conclusions are valid in the context of reinforcement learning too, e.g., Ohnishi et al. (2019); Achiam et al. (2019); Tessler et al. (2018); Uchibe & Doya (2007). Tessler et al. (2018) presents a constrained policy optimization, which uses an alternative penalty signal to guide the policy. The above safe learning approaches use a model to explicitly define constraints on it, while some (e.g., Berkenkamp et al. (2017); Cowen-Rivers et al. (2022)) use the model to explore the constraints or modify the agent's loss to penalize constraint violation, e.g., Peng et al. (2021); Ren et al. (2022). Gu et al. (2022) reviews the progress of safe RL from a real-world implementation point of view.

In this paper, we do not use mathematical models to mimic state transitions and evaluate the policy; thus, our approach is not comparable to classical model-based techniques. However, we assume some prior knowledge of the environment: the physical meaning of every element in the state-vector must be known (thus, we take a grey-box approach). Then, based on our prior knowledge,

we can select some states where we impose constraints on the policy either to completely prohibit taking some actions or to impose "soft" policy constraints that can guide learning. Our algorithm computes artificial returns to enforce these constraints during learning. Thereby, according to the proposed categorization in Garcia & Fernandez (2015), our proposed algorithm falls into constrained optimization with external knowledge. We emphasize that our approach works best when there is only partial information on the environment, not sufficient to build a proper model. Additionally, constrained RL methods in the literature assume that constraints cannot be defined in a closed form. Our contribution refutes this claim: we can define equality constraints in closed form with the help of the Neural Tangent Kernel (NTK, Jacot et al. (2018)).

This paper demonstrates constrained learning by extending the REINFORCE algorithm. REIN-FORCE is a model-free policy-based reinforcement learning algorithm. The output of the algorithm is a probability distribution, which is defined by the function approximator's $\theta$ parameters Williams (1992); Sutton & Barto (2018). Policy-based methods have shown persuasive results in various domains. Moreover, it is guaranteed that they converge at least to a local optimum Sutton et al. (1999). Policy gradient methods have many variants and extensions to improve their learning performance Zhang et al. (2021); Agarwal et al. (2020); Cheng et al. (2020). Our proposed approach can complement such variants too (see Remark 4). We develop a deterministic policy gradient algorithm (based on REINFORCE Williams (1987); Szepesvári (2010)) augmented with different types of constraints via shaping the rewards. Opposed to Altman et al. (2019), transition probabilities are not explicitly replaced with taboo states, but constraints are imposed on the policy via dynamically computing the right returns. Therefore, compared to Tessler et al. (2018), training does not rely on non-convex optimization or additional heuristics. Safety in our approach can be achieved by the careful selection of constraints (i.e., the expert knowledge).

The key element to our goals is the Neural Tangent Kernel (NTK, Jacot et al. (2018); Bietti et al. (2019); Yang & Salman (2019)), which gives us insight into the training process of a learning agent. The NTK is the scalar product of the gradients of a neural network w.r.t. its weights and biases evaluated for different pairs of states. Thus, it is a kernel function over the input space. The NTK describes how the output (in function space) of fully connected neural networks under gradient descent changes. When computing the policy evolution for commonly used loss functions, it naturally appears in the ordinary differential equation (ODE). Next, we formally define the NTK for an arbitrary neural network.

**Definition 1.** *Neural Tangent Kernel Jacot et al. (2018). Given data $x_i,\ x_j \in X \subseteq \mathbb{R}^n$, the NTK of an $n$ input one output artificial neural network $f(x,\theta):\ \mathbb{R}^n \to \mathbb{R}$, parametrized with $\theta$, is*

$$\Theta(x_i, x_j) = \left( \frac{\partial f(x_i, \theta)}{\partial \theta} \right)^T \frac{\partial f(x_j, \theta)}{\partial \theta} \in \mathbb{R}. \tag{1}$$

Using the NTK for reinforcement learning has a rather limited history. Earlier, policy iteration has been used in conjunction with NTK for learning the value function Goumiri et al. (2020). On the other hand, it was used in its analytical form as the covariance kernel of a GP (Novak et al. (2019); Rasmussen et al. (2003); Yang & Salman (2019)). The NTK-based formulation of learning enables us to evaluate how the policy output would change for arbitrary training data without actually updating the agent. This paves the way for influencing learning.

We use the NTK directly for our constrained approach. We deduce the differential equation system (which includes the NTK) from the policy gradient theorem to describe the evolution of the policy during learning. Constraints are incorporated via additional rewards for one-step-ahead policy changes. It is assumed that there are states where the agent's desired behavior is known, i.e., there is information on the environment regarding saturation, rate limits, and potential dangerous states. We call these states "safe states" and define the desired behavior in these states, which we call "policy constraints". Simply put, we explicitly define desired action probabilities (equality constraints), or prescribe a minimum action probabilities (inequality constraints) for these states. Finally, returns are computed at these safe states via convex optimization so the agent learns policies that satisfy these constraints. This method is developed for fully observable, continuous state spaces and discrete action spaces. It can directly be translated into discrete state spaces too. On the other hand, continuous action spaces would require a different approach.

The contribution of the paper is twofold. First, we analytically develop the policy evolution under gradient flow using the NTK. Second, we extend the REINFORCE algorithm with constraints.

Our variant of the REINFORCE algorithm starts converging within a few episodes if appropriate constraints are selected. The constrained extension relies on computing extra returns via convex optimization. In summary, the paper provides a practical use of the neural tangent kernel in reinforcement learning.

The paper is structured as follows. First, we present the episode-by-episode policy change of the REINFORCE algorithm (Section 2.1). Then, relying on the NTK, we deduce the policy change at unvisited states, see Section 2.2. Using the results in Section 2.2, we compute returns at arbitrary states in Section 3. We introduce equality constraints for the policy via computing safe returns by solving a system of linear equations (Section 3.1). In the same context, we can enforce inequality constraints by solving a constrained quadratic program, see Section 3.2. Additionally, we explore various ways to dynamically inject these constraints, alleviating computational complexity in high-dimensional state spaces (Section 3.3). In Section 4, we evaluate the proposed learning algorithm in different environments. Finally, we summarize the findings of this paper in Section 5.

## 2 KERNEL-BASED ANALYSIS OF THE REINFORCE ALGORITHM

In this section, the episode-by-episode learning dynamics of a policy network is analyzed and controlled in a constrained way. To this end, first we introduce the RL framework and the REINFORCE algorithm. Then, the learning dynamics of a wide and shallow neural network is analytically given. Finally, returns are calculated that force the policy to obey equality and inequality constraints at specific states.

### 2.1 REFORMULATING THE LEARNING DYNAMICS OF THE REINFORCE ALGORITHM

Reinforcement learning problems are commonly formulated via a Markov Decision Process (MDP) Sutton & Barto (2018). Similarly, the most common way of tackling safe RL is through constrained MDPs, i.e., safety is ensured via constraining the MDP: at given states, some actions that are deemed unsafe are forbidden Altman (1999); Wachi & Sui (2020).

Let the 5-tuple $(\mathcal{S}, \mathcal{A}, \underline{\underline{P}}, \mathcal{R}(s), \gamma)$ characterize an MDP. This tuple consists of the continuous state space with $n_n$ dimensions $\mathcal{S} \subseteq \mathbb{R}^{n_n}$, the discrete action space $\mathcal{A} \subset \mathbb{Z}^{n_a}$, the transition probability matrix $\underline{\underline{P}}$, the reward function $\mathcal{R}(s) \in \mathbb{R}$, and the discount factor $\gamma \in ]0, 1]$. The agent traverses the MDP following the policy $\pi(a^i \mid s, \theta)$. The policy network is parametrized with $\theta \in \mathbb{R}^{n_\theta}$ comprising of $n_\theta$ weights and biases, $a^i \in \mathbb{Z}$ is the $i^{th}$ action, $i \in 1, 2, ..., n_A$, and $s \in \mathcal{S}$ is the environment state. Throughout the paper, we denote vectors with single underline and matrices with double underlines, unless stated otherwise. For the sake of brevity, two exceptions to this convention are the parametrization of the neural network $\theta$ and the state vector $s \in \mathcal{S}$. A nomenclature is given in Appendix A.

Since policy gradient methods learn from data batches episode-by-episode, we index one episode with subscript $e$. Assuming continuous environment space and discrete action space, one episode batch (with length $n_B$) comprises of $\{\underline{s}_e, \underline{a}_e, \underline{r}_e\}$. For convenience, the states, actions, and rewards in batch $e$ are organized into columns $\underline{s}_e \in \mathbb{R}^{n_B \times n_n}$, $\underline{a}_e \in \mathbb{Z}^{n_B}$, $\underline{r}_e \in \mathbb{R}^{n_B}$, respectively. Note that the length of batches is different for each episode. For the sake of brevity, we do not explicitly indicate the episode dependency of $n_B$. Within the batch, $\underline{s}_e(k) \in \mathcal{S}$ is the $n_n$ dimensional state vector in the $k^{th}$ step of the MC trajectory, $k = 1, 2, ..., n_B$, $\underline{a}_e(k) \in \mathcal{A}$ is the action taken, and $\underline{r}_e(k) \in \mathcal{R}(s)$ is the reward obtained. In the reinforcement learning setup, the goal is maximizing the expected (discounted sum of future) rewards,

$$\underline{G}_e(k) = \sum_{\kappa=k+1}^{n_B} \gamma^{\kappa-k} \underline{r}_e(\kappa) \in \mathbb{R}, \tag{2}$$

(i.e., the return) in episode $e$, and time step $k$ of episode $e$ by iteratively updating the policy applying the policy gradient theorem Sutton et al. (1999).

In REINFORCE, the agent learns the policy directly by updating its weights $\theta_e$ using Monte-Carlo (MC) episode samples. The update rule is based on the policy gradient theorem Sutton et al. (1999)

and for the whole episode it can be written as the sum of gradients induced by the batch:

$$\theta_{e+1} = \theta_e + \alpha \sum_{k=1}^{n_B} \left( \underline{G}_e(k) \frac{\partial}{\partial \theta} \log \pi(\underline{a}_e(k) \mid \underline{s}_e(k), \theta_e) \right)^T . \tag{3}$$

In Eq. 3, $\alpha$ denotes the learning rate.

Next, we will use gradient flow ($\alpha \to 0$, Bertsekas (1997)) to analyze the policy update in a "continuous" way. We will derive how a batch of data from episode $e$ influences the policy change (i.e., the gradient of the policy update). Denote the episodic change with $\frac{\partial \pi(a_e|s_e, \theta_e)}{\partial e} \in \mathbb{R}^{n_B}$ where each row stands for the policy shift at $s_e(k)$ for action $a_e(k)$. Theorem 1 states that the policy evolution of the REINFORCE algorithm under gradient flow can be written as a nonlinear differential equation system with the help of the NTK.

**Theorem 1.** *Given batch $\{\underline{s}_e(k), \underline{a}_e(k), \underline{r}_e(k)\}_{k=1}^{n_B}$, and assuming gradient flow, the episodic policy change with the REINFORCE algorithm at the batch state-action pairs are*

$$\frac{\partial \underline{\pi}(\underline{a}_e \mid \underline{s}_e, \theta_e)}{\partial e} = \underline{\underline{\Theta}}_{\pi,e}(\underline{s}_e, \underline{s}_e) \underline{\underline{\Pi}}_e^I(\underline{s}_e, \underline{a}_e, \theta_e) \underline{G}_e,$$

*where $\underline{\underline{\Theta}}_{\pi,e}(\underline{s}_e, \underline{s}_e) \in \mathbb{R}^{n_B \times n_B}$ is the neural tangent kernel (NTK), $\underline{\underline{\Pi}}_e^I(\underline{s}_e, \underline{a}_e, \theta_e) \in \mathbb{R}^{n_B \times n_B}$ is a diagonal matrix containing the inverse policies (if they exist) at state-action pairs of batch $e$, and $\underline{G}_e \in \mathbb{R}^{n_B}$ is the vector of returns.*

The proof of the theorem employs the chain-rule on the continuous form of Eq. 3 and some linear algebra to reorganize matrix multiplications. The detailed proof of Theorem 1 alongside with some properties of the NTK-based learning dynamics is reported in Appendix B.1.

## 2.2 EVALUATING THE POLICY CHANGE FOR ARBITRARY STATES AND ACTIONS

Building on Theorem 1, we can describe how the agent's output would change at states not part of $\underline{s}_e$ too. Theorem 1 gives us a way to do so without actually training the agent with data batch $e$. Similarly, it is possible to evaluate the change of action probabilities for actions not taken. This section introduces two theorems, formally describing how to evaluate latent changes to the policy through the NTK. Theorem 2 extends Theorem 1 to multiple actions and Theorem 3 gives us the formula to evaluate the policy change at any state.

In most cases, the learning agent can perform multiple actions. Assume the agent can take $a^1, a^2, ..., a^{n_A}$ actions (the policy network has $n_A$ output channels). Previous works that deal with NTK all consider one output channel in the examples (e.g., Bradbury et al. (2018); Jacot et al. (2018); Yang & Salman (2019)) but Jacot et al. (2018) claim that a network with $n_A$ outputs can be handled as $n_A$ independent neural networks. Thus it is possible to handle multiple outputs and fit every output channel into one equation by generalizing Theorem 1. First, we introduce a new notation $\underline{\Gamma}_e(\underline{a}_e, \underline{r}_e) \in \mathbb{R}^{n_A n_B}$ for the multi-action return. This sparse vector consists of $n_A \times 1$ sized blocks with $n_A - 1$ zeros at action indexes which are not taken at $\underline{s}_e(k)$, and the original return from $\underline{G}_e(k)$ (Eq. 2) at the index of the taken action. Note that $\underline{\Gamma}_e(\underline{a}_e, \underline{r}_e)$ implicitly depends on $\underline{a}_e$ through the indexing logic and on $\underline{r}_e$ through the definition of $\underline{G}_e$. The structure of $\underline{\Gamma}_e(\underline{a}_e, \underline{r}_e)$ is easier to grasp through an example given in Remark 3 in Appendix B.2.

**Theorem 2.** *Given batch $\{\underline{s}_e(k), \underline{a}_e(k), \underline{r}_e(k)\}_{k=1}^{n_B}$, and assuming gradient flow, the episodic policy change with the REINFORCE algorithm at the batch states for an $n_A$ output policy network is*

$$\frac{\partial \underline{\pi}(\underline{a} \mid \underline{s}_e, \theta_e)}{\partial e} = \underline{\underline{\Theta}}_{\pi,e}(\underline{s}_e, \underline{s}_e) \underline{\underline{\Pi}}_e^I(\underline{s}_e, \theta_e) \underline{\Gamma}_e(\underline{a}_e, \underline{r}_e),$$

*where $\frac{\partial \pi(a|s_e, \theta_e)}{\partial e} \in \mathbb{R}^{n_A n_B}$ and $\underline{\underline{\Theta}}_{\pi,e}(\underline{s}_e, \underline{s}_e) \in \mathbb{R}^{n_A n_B \times n_A n_B}$.*

With Theorem 2, it is possible to evaluate how the policy will change at states $\underline{s}_e$ for arbitrary actions without actually updating the agent with batch data $e$. Note that the episode dependency of the actions (the $e$ subscript) has disappeared from $\underline{\pi}(\underline{a} \mid \underline{s}_e, \theta_e)$. The reason is that this vector contains the policy change for every possible action (regardless of whether or not taken) at states $\underline{s}_e$. We evaluate the log policy derivatives are for every possible action at the states in a batch.

For simplicity, we keep the previous notations, but the matrix and vector sizes are redefined for the vector output case. Detailed proof of the theorem is given in Appendix B.2.

When an agent is updated with a batch of training data, it changes the action probabilities for every state. Using the same NTK-based formulation, we derive how the policy would change at arbitrary states ($\notin \underline{s}_e$) if data batch $e$ was used for training.

**Theorem 3.** *Given batch $\{\underline{s}_e(k), \underline{a}_e(k), \underline{r}_e(k)\}_{k=1}^{n_B}$, and assuming gradient flow, the episodic policy change with the REINFORCE algorithm at any state $s_0 \notin \underline{s}_e$ is*

$$\frac{\partial \underline{\pi}(a \mid s_0, \theta_e)}{\partial e} = \underline{\underline{\vartheta}}(\underline{s}_e, s_0)\underline{\underline{\Pi}}_e^I(\underline{s}_e, \theta_e)\underline{\Gamma}_e(\underline{a}_e, \underline{r}_e) \in \mathbb{R}^{n_A},$$

*where $\underline{\underline{\vartheta}}(\underline{s}_e, s_0) = [\underline{\underline{\vartheta}}(s_1, s_0), \underline{\underline{\vartheta}}(s_2, s_0), ..., \underline{\underline{\vartheta}}(s_{n_B}, s_0)] \in \mathbb{R}^{n_A \times n_A n_B}$ is the neural tangent kernel evaluated for all $\underline{s}_e$, $s_0$ pairs.*

The main idea behind the proof of this theorem is separating the policy change into two terms: one related to the batch $\underline{\underline{\vartheta}}(\underline{s}_e, s_0)\underline{\underline{\Pi}}_e^I(\underline{s}_e, \theta_e)\underline{\Gamma}_e(\underline{a}_e, \underline{r}_e) \in \mathbb{R}^{n_A}$ and one that solely depends on the arbitrary state $s_0$: $\underline{\underline{\vartheta}}(s_0, s_0)\underline{\underline{\Pi}}_e^I(s_0, \theta_e)\underline{\Gamma}_0$. Since $s_0$ is an unvisited state it does not affect the learning. Therefore, the return associated to this state $\underline{\Gamma}_0$ is zero for every action, thus cancelled out. The full proof of Theorem 3 is described in Appendix B.3.

Although, in Theorem 3 $\underline{\Gamma}_0 = 0$, it will play a central role in defining policy constraints. Values in the place of $\underline{\Gamma}_0$ can be used to constrain the policy to obey constraints at some select states via computing arbitrary returns.

## 3 THE NTK-BASED CONSTRAINED REINFORCE ALGORITHM

In every control task, the engineer knows the desired performance metrics beforehand (i.e., how the controlled system should behave). In reinforcement learning, however, inherent dynamics of the system are black box, and the performance metrics are hidden within the rewards. On the other hand, the physical meaning of every (observable) environment state and action are most often known. With these pieces of information, it is possible to pinpoint some parts of the environment space where the desired behaviour can be described as policy constraints. That is to guide the learning of the agent (sample efficiency) and avoid potentially dangerous states (safety). For example, in a path following task Wurman et al. (2022); Szoke et al. (2022), the states are the relative position of the vehicle and the action is the steering wheel angle. We know by intuition, if the car is about to leave the road, the desired action is to steer towards the center of the road. This can be translated to constraints on the policy by saying that at those critical states the probability of taking an action that drives the vehicle towards the center of the road should be high. Similarly, we will describe the heuristics of constraint selection through two example environments given in Section 4. Thus, we use our pieces of knowledge about the environment and our intuitions to form reference policies (constraints). I.e., we specify which actions (with what probability) the agent shall take at certain states (or subsets of the state space). Note that this approach can guarantee safety to the extent of our the available information (locally) on the environment. In this section, we propose three types of constraints:

- equality constraints: the agent shall take a specific action with a fixed probability,

- inequality constraints: the agent shall take a defined action with at least a given probability,

- a methodology to dynamically prescribe constraints as learning progresses.

Enforcing these reference policies will be accomplished via computing fictitious returns. With this approach, the policy is not entirely overridden; the agent still learns using the data batch while obeying the constraints.

Denote the set of states where we have policy constraints with $\underline{s}_s = [s_{s1}, s_{s2}, ..., s_{sn_S}] \in \mathcal{S}^{n_S}$. Then, we define equality and inequality constraints as $\underline{\pi}_{ref,eq}(\underline{a}_s \mid \underline{s}_s) = \underline{c}_{eq}$ and $\underline{\pi}_{ref,ineq}(\underline{a}_s \mid \underline{s}_s) \geq \underline{c}_{ineq}$. The constraints $\underline{c}_{eq}$, $\underline{c}_{ineq}$ denote constant action probabilities. Additionally, these can be extended to dynamic constraints. In the sequel, relying on Theorem 3, we provide the mathematical deduction on how to enforce constraints during learning.

## 3.1 EQUALITY CONSTRAINTS

Let's denote the desired policy change at the constrained states $\underline{s}_s$ for the desired actions $\underline{a}_s$ as

$$\Delta\underline{\pi}(\underline{a}_s \mid \underline{s}_s, \theta_e) = \underline{\pi}_{ref,eq}(\underline{a}_s \mid \underline{s}_s) - \underline{\pi}(\underline{a}_s \mid \underline{s}_s, \theta_e), \tag{4}$$

$\Delta\underline{\pi}(\underline{a}_s \mid \underline{s}_s, \theta_e) \in \mathbb{R}^{n_S}$. According to Theorem 3, the policy change can be computed anywhere. Using Eq. 22 we can compute the unconstrained policy change at these state-action pairs. Recall, in Section 2.2 we assumed these states do not affect the learning, so we previously set the returns $\underline{\Gamma}_0 = \underline{0} \in \mathbb{R}^{n_A}$. Instead of setting these values to zero, constraints are enforced by computing returns such that they scale this policy change to match the magnitude of the desired policy change (Eq. 4). Now we intend to constrain one action at a constrained state, thus the mapping of $\Gamma$ (Remark 3) is not needed. Define safe returns $\underline{G}_s = [G_{s1}, G_{s2}, ..., G_{sn_S}]$ for the safe actions at the safe states that will scale this policy change to match the magnitude of the desired policy change (Eq. 4). To compute these safe returns, substitute Eq. 22 into Eq. 4, evaluated for the equality constrained states. We get a system of linear equations

$$\Delta\underline{\pi}(\underline{a}_s \mid \underline{s}_s, \theta_e) - \underline{\underline{\vartheta}}(\underline{s}_e, \underline{s}_s)\underline{\underline{\Pi}}_e^I(\underline{s}_e, \theta_e)\underline{G}_e = \underline{\underline{\vartheta}}(\underline{s}_s, \underline{s}_s)\underline{\underline{\Pi}}_e^I(\underline{s}_s, \theta_e)\underline{G}_s, \tag{5}$$

that has to be solved for $\underline{G}_s$. If the product $\underline{\underline{\vartheta}}(\underline{s}_s, \underline{s}_s)\underline{\underline{\Pi}}_e^I(\underline{s}_s, \theta_e)$ is invertible, Eq. 5 has a single unique solution as there are $n_S$ unknown returns and $n_S$ equations (assuming $n_S$ equality constraints). The matrix $\underline{\underline{\vartheta}}(\underline{s}_s, \underline{s}_s)$ (the NTK) is symmetric, non-negative, finite, and bounded Jacot et al. (2018), and $\underline{\underline{\Pi}}_e^I(\underline{s}_s, \theta_e)$ is a diagonal matrix with positive elements. Their product is most likely a non-singular matrix, thus invertible, but if not, its pseudoinverse can still be computed Ben-Israel & Greville (2003). Then, $\underline{G}_s$ can be found with different methods Haidar et al. (2018). This implies that the constraints need to be feasible (non-conflicting) and realizeable by the NN.

In the initial stages of learning, the difference between the reference policy and the actual one at the constrained states will be large. Therefore, high returns are needed to eliminate this difference. This also implies that the contribution of the Monte-Carlo batch to the policy update will be minor compared to the safe state-action-return tuples. In addition, large returns might cause loss of numerical stability during the learning. The returns computed from the linearized policy change might lead to policies differing from the desired one, especially if the learning rate $\alpha$ is large. When the policy obeys the constraints, the computed returns will only compensate for the policy offset the MC batch would impose at the constrainted states (see Remark 5). If there is only a minor deviation from the desired policy, these returns will be small compared to $\underline{G}_e$ and will not significantly affect the learning. In this case, the rewards from the Monte-Carlo data batch will dominate the learning. In addition, if the policy is smooth, the agent's action probabilities in the neighborhood of a constrained state will not be radically different from the constrained one if the constraints are satisfied. This also depends on the approximation capabilities and the complexity of the policy. In a continuous state space, defining constraints in a grid-based fashion is easy but computationally ineffective. Additionally, the "density" of constraints highly depends on the domain in which the agent learns.

In practice, constrained learning is implemented through augmented batches. We concatenate the safe states, actions, and computed returns with the episode batch as: $\{(\underline{s}_e, \underline{s}_s), (\underline{a}_e, \underline{a}_s), (\underline{G}_e, \underline{G}_s)\}$. Then, the agent's weights are updated with the concatenated batch with gradient ascent, Eq. 3. Therefore, the policy update is done in the same way as in the unconstrained REINFORCE algorithm. Thus, convergence properties of the learning (excluding numerical stability) are not affected.

## 3.2 INEQUALITY CONSTRAINTS

Inequality constraints can be prescribed similar to the equality ones. Instead of defining the action probabilities precisely at certain environment states, we can say an action shall be taken with at least a prescribed probability. We can write these inequality constrained reference policies as: $\underline{\pi}_{ref,ineq}(\underline{a}_s \mid \underline{s}_s) \geq \underline{c}_{ineq}$. Then, similar to Eq. 5, the inequality constraints can be written as

$$\Delta\underline{\pi}(\underline{a}_s \mid \underline{s}_s, \theta_e) - \underline{\underline{\vartheta}}(\underline{s}_e, \underline{s}_s)\underline{\underline{\Pi}}_e^I(\underline{s}_e, \theta_e)\underline{G}_e \leq \underline{\underline{\vartheta}}(\underline{s}_s, \underline{s}_s)\underline{\underline{\Pi}}_e^I(\underline{s}_s, \theta_e)\underline{G}_s. \tag{6}$$

Solving this system of inequalities can be turned into a convex quadratic programming problem. Since the original goal of reinforcement learning is learning from the collected episode batch data,

the influence of the constraints on the learning (i.e., the magnitude of $\underline{G}_s$) should be as small as possible. Therefore, the quadratic program can be formulated as:

$$\min_{G_{s_i}} \sum_{i=1}^{n_S} G_{s_i}^2 \tag{7}$$
$$s.t.\ Eq.\ 6$$

Note that the quadratic cost function is needed to penalize positive and negative returns similarly. When the constraints are already satisfied in the inequality case ($\Delta\underline{\pi}(\underline{a}_s \mid \underline{s}_s, \theta_e) = \underline{0}$), and given batch $e$, an updated policy (without augmented constrained states) would still satisfy the constraints, the optimal solution to the quadratic program would be $\underline{G}_s = \underline{0}$. In conclusion, if the inequality constraints are satisfied, the computed returns will be zero, not affecting the learning at all. The learning, in that case, follows the classical REINFORCE algorithm (opposed to Remark 5). Quadratic programming with interior point methods has has the polynomial time complexity of $\mathcal{O}(n_S^3)$, (Ye & Tse (1989)) and has to be solved after every episode. Alternatively, it is possible to relax the constraints via introducing slack variables Boyd & Vandenberghe (2004). This approach turns the optimization with hard constraints into soft ones, which can be solved even with conflicting constraints.

### 3.3 DYNAMIC CONSTRAINT SELECTION TO TACKLE LARGE STATE SPACES

The proposed algorithm can only scale well computationally in large-scale environments with some additional heuristics. Two main factors hinder scaling: i) computing the NTK for long episodes, and ii) solving Eq. 5 and Eq. 7 for many unknowns, i.e., having too many constraints. The first point can easily be resolved using mini-batches instead of full state trajectories Shen et al. (2019). The mini-batch approach effectively reduces the size of the NTK (see Definition 1) and all the other matrices used in Eq. 5 and Eq. 7. This not only means less computation but makes the optimization easier to solve.

To address the issues related to a large number of constraints, we propose a dynamic constraining approach. If a high-dimensional environment has one (or a few) state dimensions that have undesirable values, it is not only tedious to constrain every state combination but can also be computationally impracticable (see Remark 8). For example, in a lane-keeping task, lane departure is only characterized by the vehicle's lateral position, while its state vector can consist of several other components (velocity, headways, etc.). The desired action in such a situation is also evident (steer back towards the lane center). Thus, we create dynamical constraints that only use the constrained portion of the state vector and a desired action. When the agent interacts with the environment during one Monte-Carlo trajectory, it is checked in every step whether a dynamically constrained state is reached (e.g., the deviation from the lane center is higher than a threshold in the lane-keeping example). If so, that state is saved alongside the desired action and appended to the safe state batch $\underline{s}_s$ and safe action batch $\underline{a}_s$, respectively, for the current episode. Then, after the termination of the episode, the desired returns $\underline{G}_s$ are computed the same way as for the equality or inequality constraints. Mathematically, a dynamic constrained state $s_d$ has elements from $\mathcal{S}$ but $dim\{s_d\} \leq dim\{\mathcal{S}\}$. If an environment state $\underline{s}_e(k)$ has an undesirable value, a dynamic constraint can be set up (e.g., if $s_d$ is the lateral position of the vehicle, and the vehicle at state $\underline{s}_e(k)$ is departing the lane, $\|\underline{s}_e(k, \lambda)\| > s_d(\lambda)$, with $\lambda$ denoting the index of the lateral position in each vector). Finally, define this new class of desired policy (dynamic constraint) as $\underline{\pi}_{ref,d}(\underline{a}_s \mid \underline{s}_d)$, indicating that multiple constraints are handled as vectors with an underscore.

This way, the agent will only learn from its actual state trajectory and the constrained states it encountered during that episode. Thus, we do not constrain the whole, large dimensional state space, trading off safe-learning performance with computational performance.

We summarize the NTK-based constrained REINFORCE algorithm in Algorithm 1. Note that this algorithm explains the mega-batch (full Monte-Carlo trajectory) approach.

## 4 EXPERIMENTAL STUDIES

We investigate the proposed constrainted learning algorithm in three environments: Cartpole Barto et al. (1983) and Lunar lander, Brockman et al. (2016), and a highway pilot environment Bécsi

et al. (2018). The simplicity of the Cartpole environment enables us to perform an in depth analysis of the proposed algorithm. The Lunar Lander demonstrates its efficiency in higher-dimensional environment. Additionally, the highway environment is used to demonstrate dynamic constraining in a higher dimensional state space. Through these environments, it is also explained what external knowledge is needed when formulating the constraints.

---

**Algorithm 1** The NTK-based constrained REINFORCE algorithm

---

1: Define equality constraints $\underline{\pi}_{ref,eq}(\underline{a}_s \mid \underline{s}_s) = \underline{c}_{eq}$.
2: Define ineq. constraints $\underline{\pi}_{ref,ineq}(\underline{a}_s \mid \underline{s}_s) \geq \underline{c}_{ineq}$.
3: Define rules for dynamic constraints $\underline{\pi}_{ref,d}(\underline{a}_s \mid \underline{s}_d)$.
4: Initialize $e = 1$.
5: Initialize the policy network with random $\theta$ weights.
6: **while** not converged **do**
7:     Generate a MC trajectory $\{\underline{s}_e(k), \underline{a}_e(k), \underline{r}_e(k)\}$, with the current policy $\pi(\underline{a}_e(k) \mid \underline{s}_e(k), \theta_e)$.
8:     **for** the trajectory $(k = 1, 2, ..., n_B)$ **do**
9:         **if** $\underline{s}_e(k)$ violates a dynamic constraint $s_d$ **then**
10:             Append $\underline{s}_e(k)$ to $\underline{s}_s$
11:             Append the desired action to $\underline{a}_s$
12:         **end if**
13:         Compute the returns $G_e(k)$ with Eq. 2.
14:     **end for**
15:     Construct $\underline{\vartheta}(\underline{s}_e, \underline{s}_s), \underline{\vartheta}(\underline{s}_s, \underline{s}_s), \underline{\underline{\Pi}}_e^I(\underline{s}_e, \theta_e), \underline{\underline{\Pi}}_e^I(\underline{s}_s, \theta_e)$.
16:     Compute $\underline{G}_s$ with Eq. 5 **and** Eq. 7.
17:     Concatenate the MC batch and the constraints: $\{(\underline{s}_e, \underline{s}_s), (\underline{a}_e, \underline{a}_s), (\underline{G}_e, \underline{G}_s)\}$.
18:     **for** the augmented MC trajectory $(k = 1, 2, ..., n_B + n_S)$ **do**
19:         Update policy parameters with gradient ascent (Eq. 3).
20:     **end for**
21:     Increment $e$.
22: **end while**

---

The learning agent is a 2-layer deep fully-connected neural network with ReLU activations and softmax output nonlinearity and appropriate input-output sizes (i.e., 4 inputs and 2 outputs for Cartpole, 8 inputs and 6 outputs for Lunar lander, and 17 inputs and 25 outputs for the highway environment). The width of the hidden layer is 5000 neurons with bias terms. That is to comply with the assumptions in Jacot et al. (2018), i.e., a shallow and wide neural network. On the other hand, the NTK can be computed for more complex NN structures too e.g., Yang & Salman (2019). Note that the primary purpose is not finding the best function approximator, merely demonstrating the efficiency of the constrained learning. To achieve lazy learning, the learning rate is set to $\alpha = 0.0001$. The small learning rate ensures that the approximation of the policy change remains accurate.

In each environment, the constrained learning is benchmarked against a deep double Q-network (DDQN, Van Hasselt et al. (2016)), actor-critic (AC Grondman et al. (2012)), and proximal policy optimization (PPO, Schulman et al. (2017)).

Simulation results are summarized in Figure 1. In the Cartpole environment, with proper selection of the constraints, it is possible to train the agent in 5 steps, with orders of magnitude less environment interactions than for the unconstrained agents. Nevertheless, it can be attributed to the simplicity of the problem and the choice of constraints. For this environment, the constraints are chosen based on what the agent should do intuitively. The agent can only learn new policies at states which are not constrained. Thus, the final policy will be sub-optimal (if the constraints are sub-optimal). This is a trade-off between sample efficiency and optimality. The constraining makes learning much faster as it eliminates the need to explore states known to be unsafe.

In Lunar lander, the two most common reasons observed for episode failures are the lander crashing too fast into the ground and tilting over mid-flight. To this end, constraints are imposed on the lander's vertical velocity and angle (see Table 4). The agent can find a good policy within a few episodes if the constraints are set up well. On the other hand, a poor selection of constraints can harm the performance in the long run. In this environment, an on par final policy can be achieved, compared to the benchmarks in one order of magnitude less episodes.

The highway pilot environment is a higher dimensional environment, where static constraints would impose too much computational burden. On the other hand, some undesirable environment states can be isolated (e.g., driving towards the curb or towards a potential rear-end collision). If such an undesired event happens, we dynamically append a constraint featuring that specific state and a desired safety action to the learning batch. Compared to the other agents, our constrained agent starts converging faster, however, not orders of magnitude faster than in the other two environments.

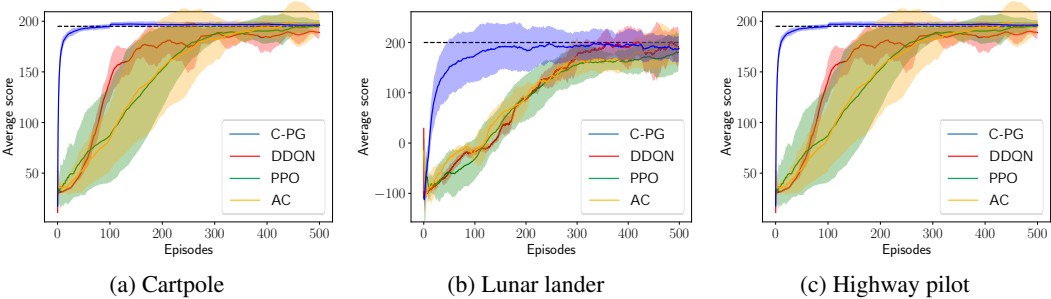

Figure 1: Average score in the benchmark environments for 5 different random seeds.

That is because the constraints cover a much smaller portion of the environment space compared to the previously discussed environments, so the constrained agent has to explore more. For the benchmark agents, the most common early termination cause was leaving the highway, followed by colliding with other vehicles. For the constrained agent, collision was the most common cause. That is because road departure was covered by a constraint, but collision was only constrained for rear-end crashes and not for colliding into vehicles on adjacent lanes.

In-depth conclusions of the results are provided in Appendix C.

## 5 CONCLUSIONS

We proposed a solution to augment the REINFORCE algorithm with equality, inequality, and dynamically changing constraints for the policy. The key to constraining is the neural tangent kernel. It enables the evaluation of the policy change without actually training the agent at arbitrary states and actions, as described by the theorems in the paper. Constrained arbitrary environment states are states with desired action probabilities based on expert knowledge of the environment. Through the NTK, desired returns are computed that approximately satisfy the prescribed constraints under gradient ascent. The actual learning is done on augmented batches, where the Monte-Carlo samples are concatenated with the constrained state-actions pairs with the precomputed returns. Simulation results suggest that constraints are satisfied after 2-3 episodes. If they are set up correctly, learning becomes extremely fast (episode-wise) while satisfying safety constraints, thus ensuring some transparency of the policy too. If the constraints are satisfied, the constrained returns become small, only slightly influencing learning from the Monte-Carlo trajectory episode batch. On the other hand, selecting suitable constraints requires expert knowledge of the environment. Therefore, the proposed algorithm is best suited for controlled physical systems where saturations and unsafe states can be pinpointed and countermeasures can be explicitly defined. However, the learning algorithm may suffer from the curse of dimensionality: in high-dimensional state spaces setting up constraints manually is tedious. To solve the optimization for large environments with the proposed polynomial time complexity algorithm, dynamically selected constraints are proposed.

As a future line of research, other variants of policy gradient methods will be analyzed. We hypothesize that more complex policy-based approaches can also be augmented with constraints using the NTK. Furthermore, feasibility analysis of constrained policies and learning constraints pose interesting lines of research.

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

## A NOMENCLATURE

| | |
|---|---|
| $\theta$ | parametrization of the function approximator (policy network), weights and biases |
| $\Theta(x_i, x_j)$ | Neural Tangent Kernel (NTK) evaluated at two data points $x_i$ and $x_j$ |
| $\mathcal{S}$ | environment state space |
| $\mathcal{A}$ | action space |
| $\underline{\underline{P}}$ | transition probability matrix |
| $\overline{\mathcal{R}}(s)$ | reward function |
| $\gamma$ | discount factor |
| $n_n$ | dimension of the environment space |
| $n_a$ | dimension of the action space |
| $\pi(\underline{a}_e(k) \mid \underline{s}_e(k), \theta_e)$ | policy in episode $e$, step $k$ |

| | |
|---|---|
| $n_\theta$ | number of weights and biases |
| $a^i$ | the $i^{th}$ discrete action |
| $s$ | environment state |
| $e$ | episode index |
| $n_B$ | data batch length (for one episode) |
| $\underline{s}_e$ | vector of visited states in episode $e$ |
| $\underline{a}_e$ | vector of actions taken in episode $e$ |
| $\underline{r}_e$ | rewards gained in episode $e$ |
| $k$ | step index in one episode, e.g., $\underline{s}_e(k)$ denotes the $k^{th}$ visited state in episode $e$ |
| $\underline{G}_e$ | vector of returns (expected discounted sum of future rewards) in episode $e$ |
| $\underline{\pi}(\underline{a}_e \mid \underline{s}_e, \theta_e)$ | policy vector in episode $e$ |
| $\underline{\underline{\Theta}}_{\pi,e}(\underline{s}_e, \underline{s}_e)$ | Neural Tangent Kernel (NTK) for state vector $s_e$ with the current policy |
| $\underline{\underline{\Pi}}^I_e(\underline{s}_e, \underline{a}_e, \theta_e)$ | diagonal matrix of inverse policies in episode $e$ |
| $\underline{\underline{\dot{\Pi}}}_{log,e}(\underline{s}_e, \underline{a}_e, \theta_e)$ | matrix of partial log policy derivatives (w.r.t. $\theta_e$) in episode $e$ |
| $\underline{\underline{\dot{\Pi}}}_e(\underline{s}_e, \underline{a}_e, \theta_e)$ | Jacobian matrix $\nabla^T_\theta \underline{\pi}(\underline{a}_e \mid \underline{s}_e, \theta_e)$ |
| $\underline{\underline{\beta}}$ | coefficient matrix |
| $\overline{\overline{V}}(\underline{\pi}(\underline{a}_e \mid \underline{s}_e, \theta_e))$ | Lyapunov function |
| $\underline{\Gamma}_e(\underline{a}_e, \underline{r}_e)$ | vector of returns considering actions not taken (with zero returns) too |
| $\underline{\underline{\Pi}}^I_e(\underline{s}_e, \theta_e)$ | diagonal matrix of inverse policies in episode $e$, for every action |
| $\underline{\underline{\dot{\Pi}}}_{log,e}(\underline{s}_e, \theta_e)$ | matrix of partial log policy derivatives (w.r.t. $\theta_e$) in episode $e$, for every action |
| $\underline{\underline{\dot{\Pi}}}_e(\underline{s}_e, \theta_e)$ | Jacobian matrix considering every possible action from visited states $\underline{s}_e$ |
| $\underline{\underline{E}}$ | identity matrix |
| $\overline{\underline{\pi}}^B(\underline{a} \mid \underline{s}_e, \theta_e)$ | averaged policy change for episode $e$ |
| $s_0$ | arbitrary environment state |
| $\underline{\underline{\vartheta}}(\underline{s}_e, s_0)$ | NTK evaluated for the episode states and an arbitrary state |
| $\underline{\Gamma}_0$ | returns for the unvisited state, for any action (zeros) |
| $\underline{s}_s$ | constrained states |
| $n_S$ | number of constraints |
| $s_d$ | dynamic constraint |
| $\underline{c}_{eq}$ | equality constraints (action probabilities) |
| $\underline{c}_{ineq}$ | inequality constraints (action probabilities) |
| $\underline{a}_s$ | desired action (by the constraint) |
| $\underline{G}_s$ | calculated returns for constraint satisfaction |
| $\Delta\underline{\pi}(\underline{a}_s \mid \underline{s}_s, \theta_e)$ | desired policy change for constraint satisfaction |
| $\underline{\pi}_{ref,eq}(\underline{a}_s \mid \underline{s}_s)$ | desired policy for equality constrained states |
| $\underline{\pi}_{ref,ineq}(\underline{a}_s \mid \underline{s}_s)$ | desired policy for inequality constrained states |
| $\underline{\underline{\vartheta}}(\underline{s}_e, \underline{s}_s)$ | NTK evaluated for the episode states and the constrained states |
| $\underline{\underline{\Pi}}^I_e(\underline{s}_s, \theta_e)$ | diagonal matrix of inverse policies evaluated at the constrained states $\underline{s}_s$ |
| $L$ | Lipschitz constant |

## B  PROOFS AND REMARKS

### B.1  PROOF OF THEOREM 1

*Proof.* Assuming very small learning rate $\alpha$, the policy update of the REINFORCE algorithm (gradient ascent) can be written in continuous form (gradient flow) Parikh & Boyd (2014):

$$\frac{d\theta_e}{de} = \sum_{k=1}^{n_B} \left( \underline{G}_e(k) \frac{\partial}{\partial \theta} log\pi(\underline{a}_e(k) \mid \underline{s}_e(k), \theta_e) \right)^T . \tag{8}$$

Furthermore, to avoid undefined logarithm, it is assumed that the evaluated policy is strictly positive. The derivative on the left hand side is a column vector with size $n_\theta$. Rewrite the differential equation in vector form as

$$\frac{d\theta_e}{de} = \underline{\dot{\underline{\Pi}}}_{log,e}(\underline{s}_e, \underline{a}_e, \theta_e)\underline{G}_e, \tag{9}$$

where $\underline{G}_e$ is evaluated for every element of $\underline{r}_e$ based on Eq. 2. The matrix $\underline{\dot{\underline{\Pi}}}_{log,e}(\underline{s}_e, \underline{a}_e, \theta_e) \in \mathbb{R}^{n_\theta \times n_B}$ comprises of partial log policy derivatives w.r.t. the parameters (weights and biases) of the policy network. Then, $\forall \theta_e(p), p \in 1, 2, ...n_\theta$:

$$\underline{\dot{\underline{\Pi}}}_{log,e}(\underline{s}_e, \underline{a}_e, \theta_e) =$$

$$\begin{bmatrix} \frac{\partial log\pi(\underline{a}_e(1)|\underline{s}_e(1),\theta_e)}{\partial \theta_e(1)} & \cdots & \frac{\partial log\pi(\underline{a}_e(n_B)|\underline{s}_e(n_B),\theta_e)}{\partial \theta_e(1)} \\ \frac{\partial log\pi(\underline{a}_e(1)|\underline{s}_e(1),\theta_e)}{\partial \theta_e(2)} & \cdots & \frac{\partial log\pi(\underline{a}_e(n_B)|\underline{s}_e(n_B),\theta_e)}{\partial \theta_e(2)} \\ \vdots & \ddots & \vdots \\ \frac{\partial log\pi(\underline{a}_e(1)|\underline{s}_e(1),\theta_e)}{\partial \theta_e(n_\theta)} & \cdots & \frac{\partial log\pi(\underline{a}_e(n_B)|\underline{s}_e(n_B),\theta_e)}{\partial \theta_e(n_\theta)} \end{bmatrix} . \tag{10}$$

Using the logarithmic derivative formula $\frac{logf(t)}{dx} = \frac{f'(t)}{f(t)}$ on every element of $\underline{\dot{\underline{\Pi}}}_{log,e}(\underline{s}_e, \underline{a}_e, \theta_e)$, the following matrix product is obtained:

$$\underline{\dot{\underline{\Pi}}}_{log,e}(\underline{s}_e, \underline{a}_e, \theta_e) =$$

$$\begin{bmatrix} \frac{\partial \pi(\underline{a}_e(1)|\underline{s}_e(1),\theta_e)}{\partial \theta_e(1)} & \cdots & \frac{\partial \pi(\underline{a}_e(n_B)|\underline{s}_e(n_B),\theta_e)}{\partial \theta_e(1)} \\ \frac{\partial \pi(\underline{a}_e(1)|\underline{s}_e(1),\theta_e)}{\partial \theta_e(2)} & \cdots & \frac{\partial \pi(\underline{a}_e(n_B)|\underline{s}_e(n_B),\theta_e)}{\partial \theta_e(2)} \\ \vdots & \ddots & \vdots \\ \frac{\partial \pi(\underline{a}_e(1)|\underline{s}_e(1),\theta_e)}{\partial \theta_e(n_\theta)} & \cdots & \frac{\partial \pi(\underline{a}_e(n_B)|\underline{s}_e(n_B),\theta_e)}{\partial \theta_e(n_\theta)} \end{bmatrix} .$$

$$\cdot \begin{bmatrix} \frac{1}{\pi(\underline{a}_e(1)|\underline{s}_e(1),\theta_e)} & \cdots & 0 \\ \vdots & \ddots & \vdots \\ 0 & \cdots & \frac{1}{\pi(\underline{a}_e(n_B)|\underline{s}_e(n_B),\theta_e)} \end{bmatrix} , \tag{11}$$

where the first matrix on the right hand side of Eq. 11 is a transposed Jacobian, i.e, $\left( \frac{\partial}{\partial \theta} \pi(\underline{a}_e \mid \underline{s}_e, \theta_e) \right)^T$. Let's denote it with $\underline{\dot{\underline{\Pi}}}_e(\underline{s}_e, \underline{a}_e, \theta_e)$. Introduce the notation $\underline{\underline{\Pi}}_e^I(\underline{s}_e, \underline{a}_e, \theta_e) \in \mathbb{R}^{n_B \times n_B}$ for the diagonal matrix of inverse policies (the second matrix on the right hand side of Eq. 11).

Substituting the introduced notations into Eq. 9, the change of the agent's weights training with the $e^{th}$ data batch is

$$\frac{d\theta_e}{de} = \underline{\dot{\underline{\Pi}}}_e(\underline{s}_e, \underline{a}_e, \theta_e)\underline{\underline{\Pi}}_e^I(\underline{s}_e, \underline{a}_e, \theta_e)\underline{G}_e. \tag{12}$$

Next, we use the chain rule to describe the learning dynamics of the policy:

$$\frac{\partial \pi(\underline{a}_e \mid \underline{s}_e, \theta_e)}{\partial e} = \frac{\partial}{\partial \theta}\pi(\underline{a}_e \mid \underline{s}_e, \theta_e)\frac{d\theta_e}{de}. \tag{13}$$

First, we extract $\frac{d\theta_e}{de}$ as in Eq. 12 and use the notation $\left( \underline{\dot{\underline{\Pi}}}_e(\underline{s}_e, \underline{a}_e, \theta_e) \right)^T = \frac{\partial}{\partial \theta}\pi(\underline{a}_e \mid \underline{s}_e, \theta_e)$:

$$\frac{\partial \pi(\underline{a}_e \mid \underline{s}_e, \theta_e)}{\partial e} =$$

$$\underline{\dot{\underline{\Pi}}}_e(\underline{s}_e, \underline{a}_e, \theta_e)^T\underline{\dot{\underline{\Pi}}}_e(\underline{s}_e, \underline{a}_e, \theta_e)\underline{\underline{\Pi}}_e^I(\underline{s}_e, \underline{a}_e, \theta_e)\underline{G}_e. \tag{14}$$

Note that $\left(\underline{\dot{\underline{\Pi}}}_e(\underline{s}_e, \underline{a}_e, \theta_e)\right)^T \underline{\dot{\underline{\Pi}}}_e(\underline{s}_e, \underline{a}_e, \theta_e) = \frac{\partial}{\partial\theta}\pi(\underline{a}_e \mid \underline{s}_e, \theta_e)\left(\frac{\partial}{\partial\theta}\pi(\underline{a}_e \mid \underline{s}_e, \theta_e)\right)^T$ is the NTK evaluated for every pair of states in $\underline{s}_e$, see Definition 1. Denote it with $\underline{\underline{\Theta}}_{\pi,e}(\underline{s}_e, \underline{s}_e) \in \mathbb{R}^{n_B \times n_B}$. Finally, the policy update due to episode batch $e$ at states $\underline{s}_e(k)$ for actions $\underline{a}_e(k)$ becomes:

$$\frac{\partial\pi(\underline{a}_e \mid \underline{s}_e, \theta_e)}{\partial e} = \underline{\underline{\Theta}}_{\pi,e}(\underline{s}_e, \underline{s}_e)\underline{\underline{\Pi}}_e^I(\underline{s}_e, \underline{a}_e, \theta_e)\underline{G}_e \tag{15}$$

$\square$

**Remark 1.** *Learning dynamics. Based on Eq. 15, the learning dynamics of the REINFORCE algorithm is a nonlinear differential equation system. If the same data batch $e_1 = e_2 = ... = e_N$ is used for training the neural network over and over again, the policy evolves as $\frac{\partial\pi(\underline{a}_e|\underline{s}_e,\theta_e)}{\partial e} = \beta\frac{1}{\underline{\pi}(\underline{a}_e|\underline{s}_e,\theta_e)}$ (with coefficient matrix $\underline{\underline{\beta}}$).*

**Remark 2.** *Stability analysis through the NTK. The linearity in NTK and policy dependency hints to use Lyapunov functions (of quadratic form) to check the convergence-stability of the nonlinear differential equation system Khalil (2002).*

*First, note that the system is stable if it converges to an equilibrium point ($\frac{\partial\pi(\underline{a}_e|\underline{s}_e,\theta_e)}{\partial e} = \underline{0}$. Since the inverse policies cannot be zero, we only have a trivial solution if the returns are zero. The NTK can only be a zero matrix if the gradients are zero in every state. Apart from the trivial solution, there can be infinitely many equilibrium points. We seek a Lyapunov function $V(\pi(\underline{a}_e \mid \underline{s}_e, \theta_e)) > 0 : \mathbb{R}^{n_B} \to \mathbb{R}$, $V(\pi(\underline{a}_e \mid \underline{s}_e, \theta_e)) = 0$ iff $\underline{\pi}(\underline{a}_e \mid \underline{s}_e, \theta_e) = \underline{0}$, and $\frac{dV(\pi(\underline{a}_e|\underline{s}_e,\theta_e))}{de} \le 0$.*

*Consider $V(\pi(\underline{a}_e \mid \underline{s}_e, \theta_e)) = \frac{1}{2}\pi(\underline{a}_e \mid \underline{s}_e, \theta_e)^T\pi(\underline{a}_e \mid \underline{s}_e, \theta_e)$ as a suitable Lyapunov function. Then*

$$\frac{dV(\pi(\underline{a}_e \mid \underline{s}_e, \theta_e))}{de} = (\pi(\underline{a}_e \mid \underline{s}_e, \theta_e))^T \underline{\underline{\Theta}}_{\pi,e}(\underline{s}_e, \underline{s}_e)\underline{\underline{\Pi}}_e^I(\underline{s}_e, \underline{a}_e, \theta_e)\underline{G}_e. \tag{16}$$

*Then, via evaluating $\frac{dV(\pi(\underline{a}_e|\underline{s}_e,\theta_e))}{de} \le 0$, we can decide whether data batch $e$ results in a locally stable training step or not.*

*Although, this remark cannot validate the convergence proof in Sutton et al. (1999), it provides an alternative way to look at the training dynamics.*

### B.2 PROOF OF THEOREM 2

*Proof.* We deduce the multi-output case by rewriting Eq. 9 and evaluate the log policy derivatives are for every possible action at the states in a batch:

$$\underline{\dot{\underline{\Pi}}}_{log,e}(\underline{s}_e, \theta_e) = \begin{bmatrix} \frac{\partial log\pi(a^1|\underline{s}_e(1),\theta_e)}{\partial\theta_e(1)} & \cdots & \frac{\partial log\pi(a^{n_A}|\underline{s}_e(n_B),\theta_e)}{\partial\theta_e(1)} \\ \frac{\partial log\pi(a^1|\underline{s}_e(1),\theta_e)}{\partial\theta_e(2)} & \cdots & \frac{\partial log\pi(a^{n_A}|\underline{s}_e(n_B),\theta_e)}{\partial\theta_e(2)} \\ \vdots & \ddots & \vdots \\ \frac{\partial log\pi(a^1|\underline{s}_e(1),\theta_e)}{\partial\theta_e(n_\theta)} & \cdots & \frac{\partial log\pi(a^{n_A}|\underline{s}_e(n_B),\theta_e)}{\partial\theta_e(n_\theta)} \end{bmatrix}, \tag{17}$$

$\underline{\dot{\underline{\Pi}}}_{log,e}(\underline{s}_e, \theta_e) \in \mathbb{R}^{n_\theta \times n_A n_B}$. The zero elements in $\underline{\Gamma}_e(\underline{a}_e, \underline{r}_e)$ will cancel out log probabilities of actions that are not taken in episode $e$, see Remark 3. Therefore, the final output $\frac{\partial\pi(\underline{a}_e|\underline{s}_e,\theta_e)}{\partial e}$ does not change. Note that, the action dependency is moved from $\underline{\dot{\underline{\Pi}}}_{log,e}(\underline{s}_e, \theta_e)$ to $\underline{\Gamma}_e(\underline{a}_e, \underline{r}_e)$. That is, because the policy is evaluated for every output channel of the policy network, but nonzero reward is given only if an action is actually taken in the MC trajectory. We continue by separating

$\underline{\dot{\underline{\Pi}}}_{log,e}(\underline{s}_e, \theta_e)$ into two matrices, in the same way as in Eq. 11.

$$\underline{\dot{\underline{\Pi}}}_{log,e}(\underline{s}_e, \theta_e) = \underline{\dot{\underline{\Pi}}}_e(\underline{s}_e, \theta_e)\underline{\underline{\Pi}}_e^I(\underline{s}_e, \theta_e) =$$

$$\begin{bmatrix} \frac{\partial \pi(a^1|\underline{s}_e(1),\theta_e)}{\partial \theta_e(1)} & \frac{\partial \pi(a^2|\underline{s}_e(1),\theta_e)}{\partial \theta_e(1)} & \cdots & \frac{\partial \pi(a^{n_A}|\underline{s}_e(n_B),\theta_e)}{\partial \theta_e(1)} \\ \frac{\partial \pi(a^1|\underline{s}_e(1),\theta_e)}{\partial \theta_e(2)} & \frac{\partial \pi(a^2|\underline{s}_e(1),\theta_e)}{\partial \theta_e(2)} & \cdots & \frac{\partial \pi(a^{n_A}|\underline{s}_e(n_B),\theta_e)}{\partial \theta_e(2)} \\ \vdots & \vdots & \ddots & \vdots \\ \frac{\partial \pi(a^1|\underline{s}_e(1),\theta_e)}{\partial \theta_e(n_\theta)} & \frac{\partial \pi(a^2|\underline{s}_e(1),\theta_e)}{\partial \theta_e(n_\theta)} & \cdots & \frac{\partial \pi(a^{n_A}|\underline{s}_e(n_B),\theta_e)}{\partial \theta_e(n_\theta)} \end{bmatrix} \cdot$$

$$\cdot \begin{bmatrix} \frac{1}{\pi(a^1|\underline{s}_e(1),\theta_e)} & & & \\ & \frac{1}{\pi(a^2|\underline{s}_e(1)\theta_e)} & & \\ & & \ddots & \\ & & & \frac{1}{\pi(a^{n_A}|\underline{s}_e(n_B),\theta_e)} \end{bmatrix}, \tag{18}$$

where the diagonalized inverse policies are denoted with $\underline{\underline{\Pi}}_e^I(\underline{s}_e, \theta_e)$. The weight change can be written as

$$\frac{d\theta_e}{de} = \underline{\dot{\underline{\Pi}}}_e(\underline{s}_e, \theta_e)\underline{\underline{\Pi}}_e^I(\underline{s}_e, \theta_e)\underline{\Gamma}_e(\underline{a}_e, \underline{r}_e). \tag{19}$$

Following the same steps as for the proof of Theorem 1, the policy change for every output channel is

$$\frac{\partial \underline{\pi}(\underline{a} \mid \underline{s}_e, \theta_e)}{\partial e} = \underline{\underline{\Theta}}_{\pi,e}(\underline{s}_e, \underline{s}_e)\underline{\underline{\Pi}}_e^I(\underline{s}_e, \theta_e)\underline{\Gamma}_e(\underline{a}_e, \underline{r}_e). \tag{20}$$

$\square$

**Remark 3.** *The structure of $\underline{\Gamma}_e(\underline{a}_e, \underline{r}_e)$ is easier demonstrated through an example. Assuming a data batch of $n_B = 4$ samples and an agent with $n_A = 3$ possible discrete actions $\underline{a}_e = [(1,0,0),(0,1,0),(0,0,1),(0,1,0)]$, and corresponding returns $\underline{G}_e = [G_e(1), G_e(2), G_e(3), G_e(4)]$,*

$$\underline{\Gamma}_e(\underline{a}_e, \underline{r}_e) = [G_e(1), 0, 0, 0, G_e(2), 0, 0, 0, G_e(3), 0, G_e(4), 0]^T. \tag{21}$$

### B.3 PROOF OF THEOREM 3

*Proof.* Since $s_0$ is not included in the learning, it does not affect the policy change. Therefore, the return associated to this state is zero for every action, $\underline{\Gamma}_0 = \underline{0} \in \mathbb{R}^{n_A}$. The NTK is based on the partial derivatives of the policy network and can be evaluated anywhere. Therefore, $\underline{\vartheta}(\underline{s}_e, s_0) = [\underline{\vartheta}(s_1, s_0), \underline{\vartheta}(s_2, s_0), ..., \underline{\vartheta}(s_{n_B}, s_0)] \in \mathbb{R}^{n_A \times n_A n_B}$ can be computed for any state. $\underline{\vartheta}(\underline{s}_e, s_0)$ consists of symmetric $n_A \times n_A$ blocks. Then the policy change at $s_0$ induced by data batch $e$ is

$$\frac{\partial \underline{\pi}(a \mid s_0, \theta_e)}{\partial e} = \underline{\vartheta}(\underline{s}_e, s_0)\underline{\underline{\Pi}}_e^I(\underline{s}_e, \theta_e)\underline{\Gamma}_e(\underline{a}_e, \underline{r}_e) +$$

$$+ \underline{\vartheta}(s_0, s_0)\underline{\underline{\Pi}}_e^I(s_0, \theta_e)\underline{\Gamma}_0. \tag{22}$$

Since $\underline{\Gamma}_0 = \underline{0} \in \mathbb{R}^{n_A}$ it cancels out the term $\underline{\vartheta}(s_0, s_0)\underline{\underline{\Pi}}_e^I(s_0, \theta_e)\underline{\Gamma}_0 \in \mathbb{R}^{n_A}$. Therefore,

$$\frac{\partial \underline{\pi}(a \mid s_0, \theta_e)}{\partial e} = \underline{\vartheta}(\underline{s}_e, s_0)\underline{\underline{\Pi}}_e^I(\underline{s}_e, \theta_e)\underline{\Gamma}_e(\underline{a}_e, \underline{r}_e) \in \mathbb{R}^{n_A}. \tag{23}$$

$\square$

### B.4 FURTHER REMARKS

**Remark 4.** *Relation to other policy gradient methods. One extension of REINFORCE is policy gradient with baseline. A baseline (typically the value function) is subtracted from the returns to reduce variance. The policy is then updated with these modified returns using the policy gradient theorem Sutton & Barto (2018). Constraints can be adapted to the policy gradient with baseline too. Since the returns at the constrained states are shaped to satisfy specific action probabilities, baselines should not be subtracted from the safe returns. Therefore, in a constrained REINFORCE with baseline, batch returns are offset by the baseline while the safe returns are not.*

**Remark 5.** *Corner cases*
*Learning with satisfied equality constraints: Assuming the constraints are already satisfied,* $\Delta \underline{\pi}(\underline{a}_s \mid \underline{s}_s, \theta_e) = \underline{0}, \forall \underline{\pi}_{ref,eq}(\underline{a}_s(\kappa) \mid \underline{s}_s(\kappa)), \kappa = 1, 2, ..., sn_S,$ *Eq. 5 simplifies to*

$$\underline{\underline{\vartheta}}(\underline{s}_e, \underline{s}_s)\underline{\underline{\Pi}}_e^I(\underline{s}_e, \theta_e)\underline{G}_e = \underline{\underline{\vartheta}}(\underline{s}_s, \underline{s}_s)\underline{\underline{\Pi}}_e^I(\underline{s}_s, \theta_e)\underline{G}_s. \tag{24}$$

*This means, the policy change imposed by batch the MC batch* $\{\underline{s}_e, a_e, \underline{r}_e\}$ *shall be compensated by the computed returns* $\underline{G}_s$*. Constraint invariance* $(\underline{G}_s = \underline{0})$ *can only be achieved, if the policy update with the MC batch makes the policy satisfy the constraints, i.e,* $\underline{\pi}_{ref,eq}(\underline{a}_s \mid \underline{s}_s) = \underline{\pi}(\underline{a}_s \mid \underline{s}_s, \theta_e) + \underline{\underline{\vartheta}}(\underline{s}_e, \underline{s}_s)\underline{\underline{\Pi}}_e^I(\underline{s}_e, \theta_e)\underline{G}_e, \forall \underline{s}_s(\kappa), \kappa = 1, 2, ..., n_S.$

***No constraints:*** *If no constraints are prescribed* $(n_S = 0)$*, we get the classic REINFORCE algorithm.*

***Fully constrainted policy:*** *If every state of the environment space is constrained,* $(n_S = \infty)$*, the agent will learn this fully constrained policy in one step (assuming no conflicting constraints). However, if a full desired policy is given via the constraints, employing a learning algorithm is pointless.*

**Remark 6.** *Smoothness of the policy: By smoothness of the policy we mean that there are no abrupt changes when looking at policies at two states close to each other. Smoothness can be defined in a Lipschitz sense for action* $a \in \mathcal{A}$ *at episode* $e$ *as*

$$\|\frac{\partial}{\partial s}\pi(a \mid s_i, \theta_e) - \frac{\partial}{\partial s}\pi(a \mid s_j, \theta_e)\|_2 \le L\|s_i - s_j\|_2, \forall s_i, s_j \in \mathcal{S}, \tag{25}$$

*with Lipschitz constant* $L$*. With the method proposed in Virmaux & Scaman (2018),* $L$ *can quickly be evaluated for deep NNs.*

**Remark 7.** *Time complexity: The critical operations are kernel evaluations and solving the linear equation system. Solving the linear equation system has the time complexity of* $\mathcal{O}(n_S^3)$ *Haidar et al. (2018). The time complexity of kernel evaluations is* $\mathcal{O}((n_B + n_S)n_S)$*. If the kernel is computed for every output channel at the batch states, time complexity would then increase to* $\mathcal{O}((n_A n_B + n_S)n_S)$*.*

**Remark 8.** *Regional constraints The equality and inequality constraints define a desired policy at specific points of the state space. However, it can be desirable to prescribe constraints to larger subsets (regions) of the state space. Constraining regions in a grid-based fashion could lead to lots of constraints that are tedious to define one by one and compute the returns (with Eq. 5 or Eq. 7). Alternatively, one can use some heuristics to limit the number of constraints defined per region. For example, in each episode, search for states within a region that have the most deviation from the desired policy and prescribe constraints to those.*

## C    DETAILED EXPERIMENTS

### C.1    CARTPOLE

The Cartpole problem is a common benchmark in control theory as it can be easily modeled as a linear time-invariant system Skogestad & Postlethwaite (2007). The goal is to balance a pole to stay upright by horizontally moving the cart. The Cartpole has four states: the position of the cart $(x)$, its velocity $(\dot{x})$, the pole angle $(\varphi)$, and the pole angular velocity $(\dot{\varphi})$. The agent in this environment can take two actions: accelerating the cart left $(a^0)$ or right $(a^1)$. In this task, the agent's goal is to balance the pole as long as possible. Reward is given for every discrete step if the pole is in vertical direction, and the episode ends if the pole falls or successfully balances for 200 steps. The pass criteria for this gym environment is reaching an average reward of 195 for 100 episodes.

For this task, we consider the constraints intuitively. If the pole is tilted too much right, the cart must move right to balance it, and vice versa. Therefore, we impose inequality constraints on the pole angle and angular velocity in a grid-based fashion. We assume the pole is tilted right if $\varphi = 0.25$, $\dot{\varphi} = 0.05$ and we must take action $a^1$ with probability $\ge 0.95$. Similarly, if $\varphi = -0.25$, $\dot{\varphi} = -0.05$ (the pole is tilted left), we must take action $a^0$ with probability $\ge 0.95$. To be invariant of the cart's position, constraints are repeated for discrete $x$ values. Inequality constraints are imposed on 18 states, see Table 3 in Appendix D. Selecting too many states to constrain slows down the computation significantly, while defining conflicting constraints can make Eq. 7 unsolvable.

With proper selection of the constraints, it is possible to train the agent in 5 steps, with orders of magnitude less environment interactions than for the unconstrained agents. Figure 2 summarizes the learning of the constrained agent compared to the other benchmarks. The agent learns the problem within a few episodes. Nevertheless, it can be attributed to the simplicity of the problem and the choice of constraints. For this environment, the constraints are chosen based on what the agent should do intuitively. Thus, once the constraints are fulfilled, the agent will be able to balance the pole. The agent can only learn new policies at states which are not constrained. Thus, the final policy will be sub-optimal (if the constraints are sub-optimal). This is a trade-off between sample efficiency and safe learning (optimality). Figure 3 depicts the section of the learned policy with the constrained states. The resulting policy is very smooth, and the constraints are fulfilled. The constraining makes learning much faster as it eliminates the need to explore states known to be unsafe.

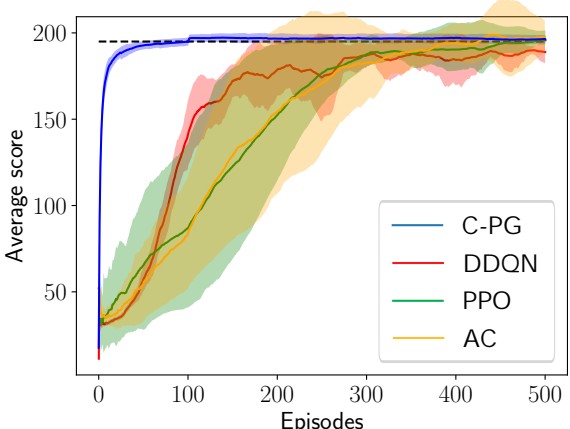

Figure 2: Average score in the Cartpole environment for 5 different random seeds.

### C.2 Demonstration of regional constraints in Cartpole

We further investigate the possibility of applying regional constraints (see Remark 8 to in a grid-based fashion. To this end, the Cartpole environment is used with the agent introduced in Section 4. For visualization purposes, we only consider the last two states of the environment: the pole angle $\varphi$, and its angular velocity $\dot{\varphi}$ at the $x = 0$, $\dot{x} = 0$ slice of the state space. First, we introduce four circular 2D disks as regional constraints, see Table 2. Next, we discretize each disk into 30 points $(x_i)$. Then, the constrained learning and select the state where the policy deviation from the

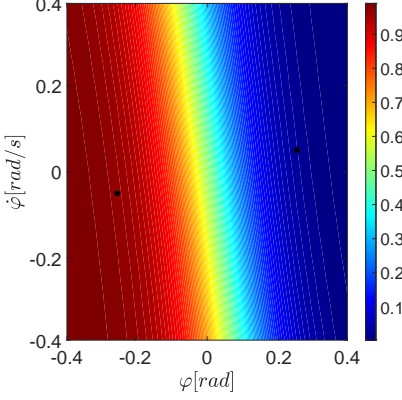

Figure 3: Probability of taking $a^0$ (push the cart left) in the Cartpole environment at $x = 0$, $\dot{x} = 0$. Dots represents the constrained states $s_{s5}$, $s_{s14}$ (Table 3)

Table 2: Constrained regions

|          | Center            | Radius (on $\varphi$-$\dot{\varphi}$) | $\pi_{ref,reg}(a^0 \mid f_s(x))$ |
|----------|-------------------|---------------------------------------|----------------------------------|
| $f_{s1}(x)$ | $[0,\ 0,\ -0.2\ -0.2]$ | 0.05 | $\geq 0.95$ |
| $f_{s2}(x)$ | $[0,\ 0,\ -0.2\ 0.2]$  | 0.05 | $\leq 0.05$ |
| $f_{s3}(x)$ | $[0,\ 0,\ 0.2\ -0.2]$  | 0.05 | $\leq 0.05$ |
| $f_{s4}(x)$ | $[0,\ 0,\ 0.2\ 0.2]$   | 0.05 | $\geq 0.95$ |

reference is maximal. Figure 4 summarizes the learning with constrained regions in the state space. The figure shows four episodes with the constrained regions (gray areas), and the corresponding constrained states (stars within the areas). In addition, the trajectory the agent traversed in that episode is shown too (black line). In constrained REINFORCE, this trajectory and the constrained states are concatenated. Thus, both the constrained states and the actual MC trajectory contribute to the learning. However, as long as the constraints are not satisfied, the rewards from the MC batch are suppressed by the large returns on the constrained states. Therefore, the two constraint selection strategies yield policies that are only slightly different from each other after learning from 9 episodes.

Note that the prescribed regional constraints in this example are arbitrary and only for demonstration purposes. In the Cartpole environment, they yield a poor policy.

## C.3 LUNAR LANDER

The goal in this 2D environment is to land a rocket on a landing pad without crashing. The agent in this environment has eight states: its horizontal and vertical coordinates $(x, y)$ and velocities $(\dot{x}, \dot{y})$, its angle $\varphi$, its angular velocity $\dot{\varphi}$ and the logical states whether the left and right legs are in contact with the ground ($l_{left}$ and $l_{right}$). The agent can choose from four actions: 0: do nothing, 1: fire the left thruster, 2: fire the main engine, and 3: fire the right thruster. The episode finishes if the lander crashes or comes to rest. Reward is given for landing successfully close to the landing pad. Crashing the rocket results in a penalty. Firing the engines (burning fuel) also results in small penalties. The pass threshold for solving this environment is an average reward of 200 for 100 episodes.

During unconstrained learning, the two most common reasons observed for episode failures are the lander crashing too fast into the ground and tilting over mid-flight. To this end, constraints are imposed on the lander's vertical velocity and angle. Based on these empirical observations, we propose inequality constraints.

Constraints are imposed to keep the lander on an ideal trajectory: as the lander comes closer to the ground, it should decelerate by firing the main engine (simulating hover slam). Intuitively, if the rocket has a too large horizontal velocity or is tilted, the side engines should be used. The proposed constraints are summarized in Table 4 (Appendix D).

With the above setup, the agent can land successfully after a few episodes. However, after 500 episodes of training, the 100 runs average reward is just below 200, see Figure 5. The results in this environment shed light on some important features of the proposed algorithm. The agent can find a good policy within a few episodes if the constraints are set up well. On the other hand, a poor selection of constraints can harm the performance in the long run. The algorithm can achieve similar performance in significantly fewer steps compared to other benchmarks, e.g., Van Hasselt et al. (2016); Gadgil et al. (2020). The unconstrained DDQN learns one order of magnitude slower. However, it can reach slightly higher average scores by the end of the training. Therefore, there is a trade-off between speeding up learning via constraining and reaching the highest possible average score. On the other hand, enforcing constraints during learning reduces oscillations and the variance from different random seeds, which is a common issue for typical benchmarks. Therefore, it is easier to know when to stop learning. With a more careful selection of constraint states (more information about the environment), the final average score of the agent could be improved, i.e., the constraints would not hamper reaching the optimal policy. This highlights one more drawback of the constrained approach: if the dimension of the environment space is large, the number of required constraints in a grid-based fashion increases significantly (i.e., the curse of dimensionality applies). Therefore, a lot more manual tuning effort is required to achieve safe and fast learning. In the next subsection, we explore the effect of dynamic constraining.

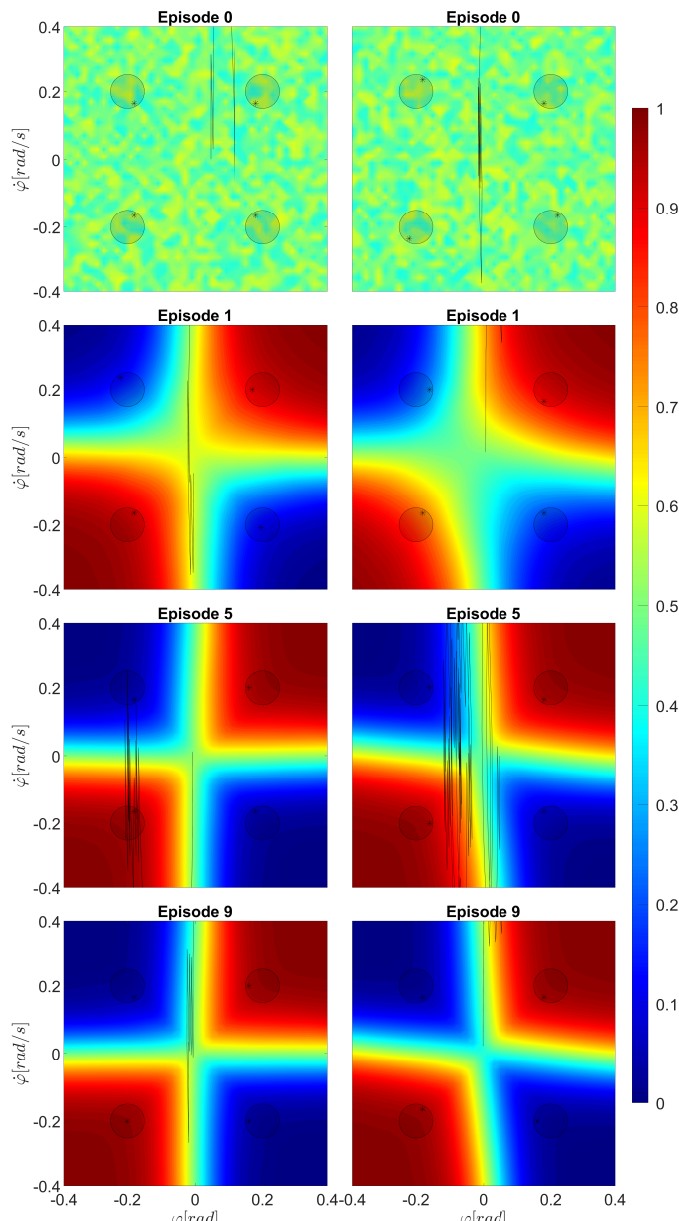

Figure 4: Probability of taking $a^0$ in the Cartpole environment at $x = 0$, $\dot{x} = 0$ under regional constraints $f_{s1}(t), ..., f_{s4}(t)$. Constrained regions are denoted with gray circles, and the actual constrained state $f_s(t_o)$ within each region by a $\star$. Black lines denote the agent's trajectory in the state space in the given episode. Plots in the left column show policy evolution with "Maximum return", plots on the right depict the "Maximum policy deviation" constraining strategy.

## C.4 HIGHWAY PILOT

This environment features a microscopic highway simulation model Bécsi et al. (2018) where a car has to navigate through dense traffic. The model features a detailed lateral motion model (based on the bicycle model). The environment has 17 continuous states, including relative distance and relative speed of the surrounding vehicles, occupancies, and vehicle states. The action space consists of 25 elements with every combination of 5 steering angle and 5 acceleration/deceleration values. The reward function consists of multiple terms such as keeping right, keeping a safe headway, and trav-

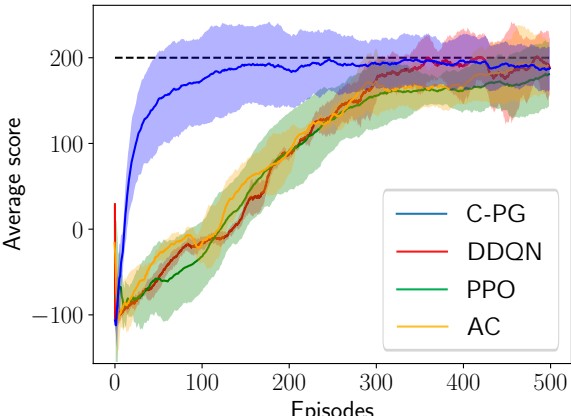

Figure 5: Average score in the Lunar lander environment for 5 different random seeds.

eling with a desired speed. The episode terminates if the vehicle travels a defined distance (success), or if it leaves the road, collides with other vehicles, or slows down too much (early termination).

The size and complexity of the environment make it hard to define full state constraints a priori. On the other hand, we can isolate some undesirable environment states derived from the early termination conditions:

- if the vehicle is approaching the (lateral) boundaries of the highway, it shall steer towards its center. Thus, $s_{d1} : y < 0.05d$, $\pi_{ref,j=2} \geq 0.95$, and $s_{d2} : y > 0.95d$, $\pi_{ref,j=22} \geq 0.95$, where $s_{d1}$, $s_{d2}$ are constraints for highway departure, $y$ is the lateral position of the vehicle and $d$ is the width of the highway (in meters, assuming three lanes). In the desired policy, the subscript $j = 2$ denotes the action with no acceleration and moderately steering left, and $j = 22$ is moderately steering right.

- If the vehicle has too small headway $h$ from the vehicle in front, it shall slow down (action $j = 11$) to avoid rear-end collision. Thus, $s_{d3} : h < 10\ m$, $\pi_{ref,j=11} \geq 0.95$.

- If the vehicle travels too slowly, it shall accelerate: $s_{d4} : v < 15\ m/s$, $\pi_{ref,j=13} \geq 0.95$. Similarly, $v$ is the longitudinal vehicle speed, and $j = 13$ is the action index for accelerating.

Constraints are also summarized in Table 5 in Appendix D. If any of these conditions are fulfilled during the episode, the actual state will be appended to the constrained state batch. See Section 3.3. We hypothesize that these constraints will significantly boost learning in the exploration phase, helping to avoid states that will certainly lead to early episode termination. Additionally, since one episode can be very long, training is done on 32-step long mini-batches.

Compared to the other agents, our constrained agent starts converging faster (Figure 6), however, not orders of magnitude faster than in the other two environments. That is because the constraints cover a much smaller portion of the environment space compared to the previously discussed environments, so the constrained agent has to explore more. For the benchmark agents, the most common early termination cause was leaving the highway, followed by colliding with other vehicles. For the constrained agent, collision was the most common cause. That is because road departure was covered by a constraint, but collision was only constrained for rear-end crashes and not for colliding into vehicles on adjacent lanes.

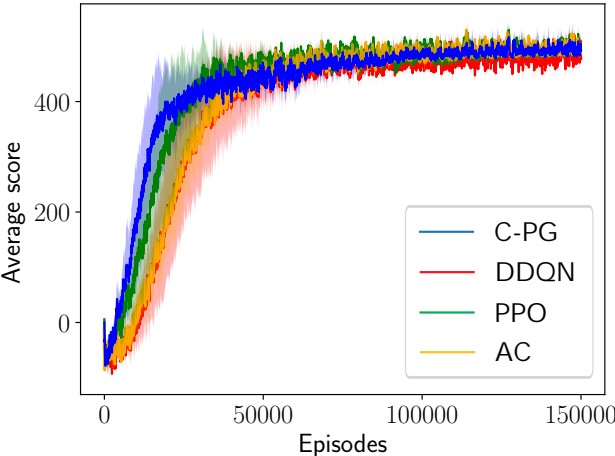

Figure 6: Average score in the Highway environment for 5 different random seeds.

## D  CONSTRAINTS IN THE SIMULATIONS

|          | $x$  | $\dot{x}$ | $\varphi$ | $\dot{\varphi}$ | $\pi_{ref}(a^0 \mid s_{s_i})$ |
|----------|------|-----------|-----------|-----------------|-------------------------------|
| $s_{s1}$  | $-2$   | 0 | 0.25   | 0.05   | $\leq 0.05$ |
| $s_{s2}$  | $-1.5$ | 0 | 0.25   | 0.05   | $\leq 0.05$ |
| $s_{s3}$  | $-1$   | 0 | 0.25   | 0.05   | $\leq 0.05$ |
| $s_{s4}$  | $-0.5$ | 0 | 0.25   | 0.05   | $\leq 0.05$ |
| $s_{s5}$  | $0$    | 0 | 0.25   | 0.05   | $\leq 0.05$ |
| $s_{s6}$  | $0.5$  | 0 | 0.25   | 0.05   | $\leq 0.05$ |
| $s_{s7}$  | $1$    | 0 | 0.25   | 0.05   | $\leq 0.05$ |
| $s_{s8}$  | $1.5$  | 0 | 0.25   | 0.05   | $\leq 0.05$ |
| $s_{s9}$  | $2$    | 0 | 0.25   | 0.05   | $\leq 0.05$ |
| $s_{s10}$ | $-2$   | 0 | $-0.25$ | $-0.05$ | $\leq 0.95$ |
| $s_{s11}$ | $-1.5$ | 0 | $-0.25$ | $-0.05$ | $\geq 0.95$ |
| $s_{s12}$ | $-1$   | 0 | $-0.25$ | $-0.05$ | $\geq 0.95$ |
| $s_{s13}$ | $-0.5$ | 0 | $-0.25$ | $-0.05$ | $\geq 0.95$ |
| $s_{s14}$ | $0$    | 0 | $-0.25$ | $-0.05$ | $\geq 0.95$ |
| $s_{s15}$ | $0.5$  | 0 | $-0.25$ | $-0.05$ | $\geq 0.95$ |
| $s_{s16}$ | $1$    | 0 | $-0.25$ | $-0.05$ | $\geq 0.95$ |
| $s_{s17}$ | $1.5$  | 0 | $-0.25$ | $-0.05$ | $\geq 0.95$ |
| $s_{s18}$ | $2$    | 0 | $-0.25$ | $-0.05$ | $\geq 0.95$ |

Table 3: Constrained states in Cartpole

|          | $x$  | $y$ | $\dot{y}$ | $\varphi$ | $l$ | $\underline{\pi}_{ref} \geq 0.95$ |
|----------|------|-----|-----------|-----------|-----|-----------------------------------|
| $s_{s1}$ | 0    | 0   | 0         | 0         | 1   | $j = 0$                           |
| $s_{s2}$ | 0    | 0.9 | $-1$      | 0         | 0   | $j = 2$                           |
| $s_{s3}$ | 0    | 0.5 | $-0.75$   | 0         | 0   | $j = 2$                           |
| $s_{s4}$ | 0    | 0.2 | $-0.5$    | 0         | 0   | $j = 2$                           |
| $s_{s5}$ | 0    | 0.1 | $-0.5$    | 0         | 0   | $j = 2$                           |
| $s_{s6}$ | 0    | 1   | 0         | $-0.25$   | 0   | $j = 1$                           |
| $s_{s7}$ | 0    | 1   | 0         | 0.25      | 0   | $j = 3$                           |
| $s_{s8}$ | 0    | 0.5 | 0         | $-0.25$   | 0   | $j = 1$                           |
| $s_{s9}$ | 0    | 0.5 | 0         | 0.25      | 0   | $j = 3$                           |
| $s_{s10}$ | 0.3 | 1.3 | 0.1       | 0         | 0   | $j = 1$                           |
| $s_{s11}$ | $-0.3$ | 1.3 | $-0.1$  | 0         | 0   | $j = 3$                           |

Table 4: Constrained states in Lunar lander

|          | State                  | Threshold   | $\underline{\pi}_{ref} \geq 0.95$ |
|----------|------------------------|-------------|-----------------------------------|
| $s_{d1}$ | $y$ (lateral position) | $< 0.05d$   | $j = 2$ (steer left)              |
| $s_{d2}$ | $y$ (lateral position) | $< 0.95d$   | $j = 22$ (steer right)            |
| $s_{d3}$ | $h$ (headway)          | $< 10\ m$   | $j = 11$ (decelerate)             |
| $s_{d4}$ | $v$ (vehicle speed)    | $< 15\ m/s$ | $j = 13$ (accelerate)             |

Table 5: Constrained states in the Highway environment

