# OpenReview forum: "A Neural Tangent Kernel Approach for Constrained Policy Gradient Reinforcement Learning"
_ICLR.cc/2024/Conference — ICLR 2024 Conference Withdrawn Submission_

### Official Review · Reviewer_RmLh · 2023-10-14

**Soundness:** 1 poor
**Presentation:** 2 fair
**Contribution:** 1 poor
**Rating:** 3
**Confidence:** 3

**Summary:**

The paper presents a constraint policy gradient method based on the clasical REINFORCE algorithm and on neural tangent kernels (NTK) which allows to evaluate the policy change in abritrary states. The authors assume partial knowledge of the prototypical behavior of the agent on certain states so as to guarantee safe behavior. They deal with equality, inequality and dynamic constraints on the policy and present simulation results on 3 simulated environments.

**Strengths:**

The main strength and originality of the paper is the novel use of NTK which allows to evaluate the policy change on arbitrary state-action pairs without actually training on those pairs. The problem that the paper deal with, i.e., constraint reinforcement learning is an important problem in the RL space.

**Weaknesses:**

I think that the authors should make it more clear that the constraints are imposed on the policy itself. I was initially under the impression that the paper dealt with a problem similar to the CPO paper by Achiam or the paper by Tessler.

With that being said, I am not convinced about how meaningfull it is to impose constraints (especially equality constraints) on the policy. Is there any other paper dealing with such a formulation?

The math of the paper is hard to follow. In the main text, the authors make references to equations or remarks in the appendix and given some of my concerns bellow I am not convienced about the correctness of the derived theoretical results.

I found the experimental results insuffecient and lacking details. What are the constraints imposed on the policy? Also,
the baselines you are comparing against (PPO, AC and DDQN) do not take any constraints into account. So how come a constrained policy performs better than an unconstraint one? Intutively, it should be the other way around. Further justifications are needed for this.

**Questions:**

Top of page 2: "We emphasize that our approach works best when there is only partial information on the environment, not sufficient to build a proper model" -> What if there is full knowledge of the environment and we can built a model? Would this add any value to your method?

Top of page 2: "constrained RL methods in the literature assume that constraints cannot be defined in a closed form" -> Please add a reference for that claim.

Botom of page 2: "we extend the REINFORCE algorithm with constraints." I do not have any spesific paper in mind, but I'd assume that the constrainted REINFORCE has been studied elsewhere. Perhaps you might want to elaborate a bit on the details of that contribution.

In the statement of theorem 1: What do you mean by "inverse policies (if they exist)"? What if they do not exist? Also, what does the partial derivative wrt \epsilon mean? Based on the notation on page 2, \epsilon is a subscript for the episode. However, in theorem 2 you mention "data batch \epsilon". So what does epsilon denote? Is it an index or a batch of data.

In eq. 4, what is the "desired policy change" and why are you referring to "constrained states"? My undestanding is that it's the policy that's being constrained not the state space.

Do \Gamma and G refer to the same quantity?

Regarding eq.4, 5 and 22: I cannot understand how eq. 5 can be derived from 4 by substituting eq. 22. If we take the second term in the LHS of eq. 5 to the RHS, then the RHS will be identical to the RHS of eq. 22. Which would imply $\Delta\pi = \frac{\partial\pi}{\partial\epsilon}$. If that is indeed the definition of the partial derivative wrt $\epsilon$ then eq. 4,5,22 are trivially the same.

In Algorithm 1, line 3: How are the rules of the dynamic constraint defined?

---

### Official Review · Reviewer_4Ecu · 2023-10-29

**Soundness:** 2 fair
**Presentation:** 3 good
**Contribution:** 3 good
**Rating:** 5
**Confidence:** 4

**Summary:**

This paper presents a method to incorporate different types of constraints on the policy output of a neural network agent trained with the REINFORCE algorithm, using the neural tangent kernel (NTK) to evaluate and control the policy change at arbitrary states and actions. The paper claims that their method can improve sample efficiency, safety, and transparency of reinforcement learning in continuous state spaces and discrete action spaces, where some prior knowledge of the environment is available. The paper demonstrates their method on three environments: Cartpole, Lunar lander, and highway pilot.

**Strengths:**

- The paper provides a novel and elegant use of the NTK for constrained policy optimization, which has not been explored much in the literature.
- The paper derives rigorous and clear theorems that relate the NTK to the policy change under gradient flow and to the computation of constrained returns.
- The paper shows empirical results that illustrate the benefits of constrained learning in terms of speed, safety, and transparency in different environments.
- The paper proposes a dynamic constraint selection scheme that can cope with high-dimensional state spaces by exploiting local information.

**Weaknesses:**

- The paper relies on expert knowledge to define suitable constraints for each environment. It would be desirable to have some guidelines or criteria for selecting appropriate constraints or learning them from data or feedback.
- The paper does not provide much theoretical analysis or guarantees on the convergence, optimality, or robustness of the constrained learning algorithm. It would be valuable to have some insights into how the choice of constraints affects these properties and how they compare with unconstrained methods.
- The paper does not discuss much about the computational complexity or scalability of their method. It would be helpful to report some statistics on the running time, memory usage, or number of iterations required by their method versus baseline algorithms.
- The baseline algorithms selected in the paper are not very fair comparisons as the expert knowledge is completely not used.
- The code is not provided for reproduction.

**Questions:**

- The baseline algorithms do not leverage the expert knowledge/constraints, which makes the comparison not very fair. Maybe the proposed algorithm should be compared with other constrained RL methods? Some other feasible approaches are, to use the expert knowledge/constraints to pre-train the policy network, or to incorporate the constraints into the policy network using methods proposed in some of Zico Kolter’s work.
- Is it possible to extend the method to algorithms other than REINFORCE?
- How does the method deal with non-smooth policies or non-linear activation functions? Does it require any modifications or approximations?

---

### Official Review · Reviewer_P7MJ · 2023-10-30

**Soundness:** 3 good
**Presentation:** 3 good
**Contribution:** 3 good
**Rating:** 6
**Confidence:** 3

**Summary:**

This paper presents a constrained policy gradient method by augmenting the traditional REINFORCE algorithm. First, the authors analyze how the agent’s policy changes if a new data batch is applied, leading to a nonlinear differential equation system in continuous time (gradient flow). This description of learning dynamics is connected to the neural tangent kernel (NTK). Next, the authors introduce constraints for action probabilities based on the assumption that there are some environment states where people know how the agent should behave. Then, the authors augment the training batch with these states and compute fictitious rewards for them, making the policy obey the constraints with the help of the NTK-based formulation. The authors also provide simulation results to demonstrate that adding constraints to the learning can improve learning in terms of speed and transparency reasonably.

**Strengths:**

1.	The studied problem, i.e., policy gradient RL with NTK and constraints, is interesting and well-motivated.
2.	The authors develop a rigorous analysis of the gradient flows for the augmented REINFORCE algorithm with NTK and constraints
3.	The authors provide experiments to demonstrate the effectiveness of their algorithm and their theoretical claims.

**Weaknesses:**

1.	What is the assumption that the authors need for the NTK setting, e.g., what size the neural network should be of?
2.	Is there any theoretical performance guarantee for the proposed algorithm? For example, is there any convergence or sample complexity guarantee?
3.	In experiments, the authors use a 2-layer deep fully-connected neural network with 5000-neuron width. Is this setup consistent with the assumption needed in theory?

**Questions:**

Please see the weaknesses above.

---

### Official Review · Reviewer_xmYU · 2023-11-01

**Soundness:** 3 good
**Presentation:** 2 fair
**Contribution:** 3 good
**Rating:** 5
**Confidence:** 3

**Summary:**

The paper implements REINFORCE algorithm with constraints using NTK. The constraints are dealt by safe rewards. The safe rewards can be computed using NTK.

**Strengths:**

The paper tackles important problem, i.e. constrained RL problem, which is known to be difficult. Directly computing the NTK to calculate the "safe rewards" seems to novel and practical approach.

**Weaknesses:**

- The paper does not guarantee theoretical results whether the algorithm convergence or what the stationary points are. Even though Remark 2 states about the stability, even for softmax policy, the quadratic Lyapunov function is not a popular choice. Furthermore, simply adding the safe reward, does not theoretically guarantee whether the constraints will be met.

- The method to calculate "safe rewards" for equality and inequality constraints seems to be quite heuristic and not really intuitive. How does equation (4) and (22) relate to equation (5)? Do we subsitute $\underline{\Gamma}_0$ with $\underline{G}_s$ in (22)? Please provide more explanations on (22).

- Implementation details of NTK are omitted, which is non-trivial, and needs more discussion. I would strongly recommend to submit the codes for the implementation.

- The paper has overload of notations which makes difficult to digest the overall meanings of the equations. It would be helpful if the authors remind the notations several times.

**Questions:**

- In equation (8), why do we have transpose? What is the dimension of $\frac{\partial}{\partial \theta} \log \pi (\underline{a}_e(k) \mid \underline{s}_e(k),\theta_e ) $?

- As for Theorem 2, how can we can treat $n_A$ outputs independent? The sum of probability should be equal to one, hence there are dependeny among the outputs.

- It would be better to introduce mathematical definition of $\underline{\Gamma}_e$ as in equation (21).

### Minor Comments

- In page 2, the sentence " This paves the way for influencing learning" is not clear.

- What is the meaning for the notation underline?

- In the Nomenclature section, it would be really helpful if vector or matrices dimensions are added, e.g., $ \underline{G}_e \in \mathbb{R}^{n\times m}$.

- Using the same notation for continuous time and discrete time ($e$) incurs confusion throughout the paper.